# Enhancing Complex Question Answering over Knowledge Graphs through Evidence Pattern Retrieval

## ABSTRACT

Information retrieval (IR) methods for KGQA consist of two stages: subgraph extraction and answer reasoning. We argue current subgraph extraction methods underestimate the importance of structural dependencies among evidence facts. We propose Evidence Pattern Retrieval (EPR) to explicitly model the structural dependencies during subgraph extraction. We implement EPR by indexing the atomic adjacency pattern of resource pairs. Given a question, we perform dense retrieval to obtain atomic patterns formed by resource pairs. We then enumerate their combinations to construct candidate evidence patterns. These evidence patterns are scored using a neural model, and the best one is selected to extract a subgraph for downstream answer reasoning. Experimental results demonstrate that the EPR-based approach has significantly improved the F1 scores of IR-KGQA methods by over 10 points on ComplexWebQuestions and achieves competitive performance on WebQuestionsSP.

## CCS CONCEPTS

• **Information systems** → **Question answering**.

## KEYWORDS

knowledge graph, question answering, information retrieval, evidence pattern

**ACM Reference Format:**
Anonymous Author(s). 2023. Enhancing Complex Question Answering over Knowledge Graphs through Evidence Pattern Retrieval. In *Proceedings of The ACM Web Conference (TheWebConf '24).* ACM, New York, NY, USA, 9 pages. https://doi.org/XXXXXXX.XXXXXXX

## 1 INTRODUCTION

With the rapid progress of large-scale knowledge graphs (KGs), there is an increasing demand for convenient and precise access to information stored within these KGs. Question answering over knowledge graphs (KGQA), i.e., the task of answering factual questions using knowledge graph facts, has gained significant attention [21]. The mainstream KGQA approaches can be roughly classified into semantic parsing (SP) approaches and information retrieval (IR) approaches. SP approaches parse natural language questions into executable queries and IR approaches retrieve answers through neural models. In recent years, KGQA research has focused on

solving complex questions that require multi-hop reasoning. SP approaches have made significant progress in solving complex questions [9, 27, 35, 37]. However, their performance relies on extensive gold question-query pairs. Without the labor-intensive annotation of gold queries, these SP approaches may either be untrainable or exhibit a significant decrease in performance. The development of in-context learning techniques has promoted the practical methods for few-shot KGQA. However, current studies [22, 23, 25, 33] indicate that these methods still lag behind the performance of the best SP and IR methods. IR approaches avoid labor-intensive query annotation by collecting the neighboring information of the topic entities (i.e., entities mentioned by the question). Because the scale of the entire KG does not support efficient training and retrieval, IR systems first extract a subgraph from the KG and then only process the information on this subgraph. Therefore, subgraph extraction greatly impacts the performance of IR approaches. While there has been some effort in extracting high-quality subgraphs [18, 28, 38], the performance of IR approaches still lags far behind that of SP approaches on complex questions.

We find that current IR studies primarily focus on how to obtain the answer(s) but pay insufficient attention to non-answer parts in the extracted subgraph. An entity can be considered an answer to the input question only if specific facts surrounding it serve as corresponding evidence. Whether a fact acts as evidence depends not only on its content but also on how it describes the topic entities and answers, specifically the structural dependencies among the relevant facts. Although current studies have considered iteratively selecting facts with question-related relations during subgraph extraction [18, 38] or the downstream reasoning [15] stage, their approaches do not provide an explicit semantic representation of the structural dependencies. We find that they sometimes include more noises in the retrieval results which may hurt performance. As illustrated in Figure 1, the facts about the noisy answer Austria have very similar relations to the evidence facts, which may cause ranking errors.

In this paper, we formulate the structural dependencies as *evidence pattern* and propose *evidence pattern retrieval* (EPR) to reduce noises in subgraph extraction. Specifically, evidence pattern models how necessary resources (topic entities and relations) are connected to support a knowledge graph node as an answer to the question. Figure 2 illustrates the corresponding evidence pattern for the question in Figure 1. Section 3 will introduce the concept of evidence pattern in detail. We train a neural model to retrieve possible atomic patterns for the input question. We obtain candidate EPs by enumerating combinations of atomic patterns and train a scoring model to select the best EP for subgraph extraction. We incorporate EPR into existing answer reasoning methods and conduct experiments on ComplexWebQuestions [30] and WebQuestionsSP [36], the two most widely used datasets in IR-KGQA evaluations. Experimental results show that evidence pattern retrieval greatly enhances

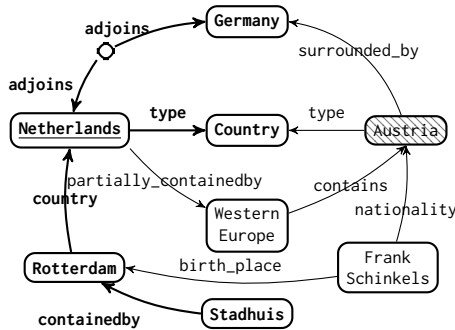

**Figure 1: Facts about question "*What country, containing Stahuis, does Germany border?*". The evidence facts are bolded., the node of the correct answer `Netherlands` is underlined, and the noisy answer `Austria` is shaded. Austria is a noisy answer since it does not contain Stahuis, but the relations on the paths between them express similar meanings and confuse the answer reasoning model.**

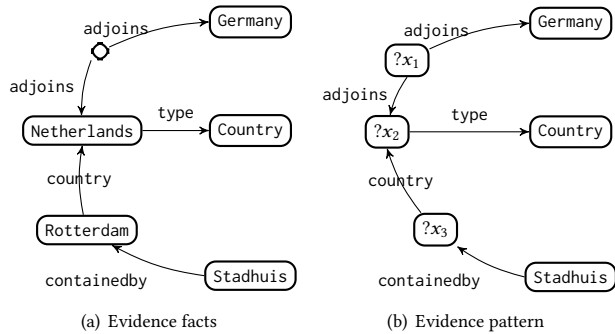

(a) Evidence facts          (b) Evidence pattern

**Figure 2: The evidence facts (a) of question "*What country, containing Stahuis, does Germany border?*" and the corresponding pattern (b).**

the performance of IR methods on ComplexWebQuestions with competitive performance on WebQuestionsSP.

The rest of this paper is organized as follows. The next section summarizes related work. The third section formulates the IR-KGQA task and EPR. The fourth section introduces the implementation of EPR in detail. The fifth section presents the experimental results with analysis. The last section concludes this paper.

## 2 RELATED WORK

### 2.1 KGQA Benchmarks

The research community has proposed lots of question-answering datasets [2, 4, 5, 32] over large-scale open-domain knowledge graphs over the past decades. Recently, researchers have begun to pay more attention to complex questions that require reasoning on multi-hop evidence [16, 30]. Some datasets follow WebQuestions [2] to collect questions first and then annotate them. Bao et al. [1] uses question-answer (QA) pairs collected from WebQuestions [2] together with manually labeled QA pairs to construct ComplexQuestions. Talmor and Berant [30] proposes ComplexWebQuestions, where complex questions are generated by composing simpler questions in WebQuestionsSP [36], the cleaned version of WebQuestions that is rephrased by AMT workers. Some other studies have taken a different approach, generating queries first and then providing corresponding natural language questions for those queries. MetaQA [8] and LC-QuAD [31] generate questions with several tens of pre-defined templates. LC-QuAD 2.0 [12] extends the framework of LC-QuAD via revised templates and crowd-sourcing tasks. Gu et al. [14] constructs GrailQA with crowd-powered paraphrasing on manually annotated canonical questions for evaluating KGQA in three different levels of generalization. Cao et al. [6] introduces KQA Pro with a compositional programming language KoPL to represent the reasoning process explicitly.

### 2.2 Information Retrieval Methods for KGQA

The mainstream solutions of KGQA first find the entities in the question (i.e., topic entities), and then search for the answers around these entities [21]. Information retrieval methods for KGQA (IR-KGQA methods) use neural models to directly score candidate answers and determine an answer set based on a score threshold. The earlier IR-KGQA methods mainly focus on simple questions that only require a 1-hop reasoning [4, 11, 24, 34]. For complex questions, IR-KGQA methods limit the search space by considering a subgraph of the entire KG. Saxena et al. [26] models the QA task via link prediction models. It restricts KG to 2-hops neighbors of topic entities and prunes the relations according to the training data. GraftNet [29] heuristically extracts a subgraph with personalized PageRank scores computed on the neighborhoods of topic entities. It ranks the extracted nodes via a graph convolutional network to predict the answers. Recent studies learn to extract question-specific subgraphs through neural models. PullNet [28] proposes a framework to iteratively construct question-specific subgraphs. It trains a GCN to identify subgraph nodes that should be "pulled" and predict the answers following GraftNet. He et al. [15] enhances neural state machine with a teacher network for providing intermediate supervision signals. Zhang et al. [38] proposes a trainable subgraph retriever to reduce the reasoning bias. It expands relation paths via a sequential decision process. UniKGQA [18] unifies subgraph extraction and answer reasoning via a semantic matching module for matching question-related relations and a propagation module to propagate the matching information.

The recent advancements in subgraph extraction have highlighted the significance of extracting question-related facts. However, these efforts primarily concentrate on specifying individual facts or relations, neglecting the crucial structural dependencies that enable these facts to support the answers.

### 2.3 Semantic Parsing Methods for KGQA

Semantic parsing methods for KGQA (SP-KGQA methods), distinct from IR-KGQA methods, represent another major category of mainstream methods that parse questions into executable queries to obtain the answer(s). Classic SP-KGQA methods depend on syntactic parsing of questions, which can be challenging when dealing with

heterogeneity between questions and query expressions [21]. Current representative SP-KGQA methods utilize pre-trained Seq2Seq models to generate queries and use retrieved information (e.g., question-relevant context or intermediate results) to augment the input [9, 35, 37] or restrict the decoding space [13, 17, 27]. For examples, Das et al. [9] proposes the idea of case-based reasoning to leverage query structures from similar questions, significantly improving performance on complex questions. Ye et al. [35] uses heuristically retrieved candidate queries as auxiliary information to augment the generation. Shu et al. [27] proposes multi-grained retrieval to restrict the decoding process with relevant KG context. These methods require fine-tuning Seq2Seq models with gold query annotations. Yu et al. [37] proposes a joint decoding approach that can work even in situations with only answer annotations but with a noticeable drop in performance. Cao et al. [7] leverages the annotation on the rich-resourced scenarios to improve the performance on scenarios that lack query annotations.

While SP-KGQA methods perform well on the benchmarks, their performance heavily relies on gold query annotations. The recent advancement in in-context learning enables practical few-shot KGQA methods that only require a limited number of query annotations. However, their strict few-shot performance still lags behind the current SOTA methods [22, 23, 25].

## 3 TASK FORMULATION

In this paper, we formalize a knowledge graph (KG) $\mathcal{G}$ as a set of triplet facts to describe entities $E$ via their relations $R$, i.e., $\mathcal{G} \subseteq E \times R \times E$. For a given question $q$, the KGQA task is to obtain the answer(s) $A_q \subseteq E$ according to $\mathcal{G}$. The information retrieval (IR) KGQA methods maximize the probability $\text{Pr}(e \in A_q)$ to distinguish $A_q$ from other entities. Since exploring the entire KG is computationally expensive, the majority of practical IR-KGQA methods operate under the assumption that a question-relevant subgraph $SG_q^* \subseteq \mathcal{G}$ exists, where $\text{Pr}(e \in A_q|q, \mathcal{G}) = \text{Pr}(e \in A_q|q, SG_q^*)$. Therefore, IR-KGQA methods are divided into two stages in practice. The first stage extracts a question-relevant subgraph to approximate $SG_q^* \subseteq \mathcal{G}$, and the second stage maximizes the probability of answers. We denote them as *subgraph extraction* and *answer reasoning* respectively. From the probabilistic perspective, subgraph extraction models a latent distribution $\text{Pr}_\phi$ and maximizes $\text{Pr}_\phi(SG_q^*)$, answer reasoning models a distribution $\text{Pr}_\psi$ on the extracted subgraph to approximate $\text{Pr}$. We formulate them as follows:

$$SG_q = \text{Ext}(\mathcal{G}, q, \text{Pr}_\phi),$$
$$A_{pred} = \left\{ e \in SG_q \mid \text{Pr}_\psi(e \in A_q|q, SG_q) > \theta \right\}, \quad (1)$$

where $\text{Ext}$ denotes a subgraph extractor, $A_{pred}$ denotes the predicted answers, $\theta$ is a confidence threshold for determining the answer set.

This paper focuses on subgraph extraction. We argue that including noisy facts in subgraph extraction will affect the consequent reasoning stage, i.e., the ideal subgraph $SG_q^*$ for question $q$ is formed by a minimal set of *evidence facts*. Given an appropriate similarity measure $sim$ over graphs and questions, $SG_q^*$ should be the most similar subgraph to question $q$, i.e.,

$$\underset{SG \subset \mathcal{G}}{\arg\max} \text{Pr}_\phi(SG|q) = \underset{SG \subset \mathcal{G}}{\arg\max} sim(SG, q). \quad (2)$$

We assume that the similarity is determined by the adjacency structure over topic entities. We model it as the *evidence pattern* $pat(SG, q)$ of $SG$. Specifically, the evidence pattern is a variable substitution of $SG$, where all entities not appearing in the question $q$ are replaced by variable symbols, as illustrated in Figure 2. Therefore, the task of subgraph extraction can be achieved through *evidence pattern retrieval*. The extraction target can be formulated as follows:

$$\underset{SG \subset \mathcal{G}}{\arg\max} sim(pat(SG, q), q). \quad (3)$$

Specifically, after retrieving the most appropriate evidence pattern $P$, we instantiate $P$ by the maximum graph that satisfies $P$, i.e., the subgraph with maximum entities. This subgraph of KG will be provided to the consequent answer reasoning.

## 4 EVIDENCE PATTERN RETRIEVAL

For a specific knowledge graph $\mathcal{G}$. The retrieval space can be denoted as $\{pat(SG) \mid SG \subseteq \mathcal{G}\}$. Given that the space exceeds the scale of manageable storage, our approach analyzes EP at the granularity of the atomic adjacency structure among entities and relations, denoted as *atomic patterns* (APs). Each AP consists of a pair of adjacent resources and defines their connection structure. Specifically, AP includes the directed connection of entity-relation pairs (denoted as ER-APs) and relation-relation pairs (denoted as RR-APs). As illustrated in Figure 3, the EP of a question can be covered by corresponding APs. We achieve EPR through the indexing and retrieval of atomic patterns, encompassing all conceivable EP instances.

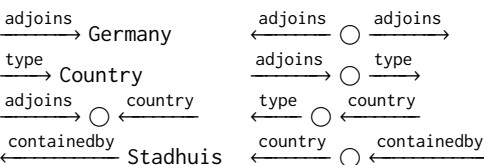

**Figure 3: Atomic patterns appeared on the evidence pattern in Figure 2.**

For an input question, we use a fine-tuned bi-encoder model to retrieve candidate APs and enumerate the possible EPs with an iterative pattern expansion algorithm. If there are multiple candidate EPs, we score them via a ranking model to select the best EP.

### 4.1 Atomic Pattern Retrieval

For an input question, the space of possible ER-APs is restricted by the topic entities, but the space of possible RR-APs includes all adjacent relation pairs in KG. Therefore, we have to build a fast index for retrieving candidate RR-APs.

We follow the dense retrieval fashion [20] to build Faiss [19] indexes of RR-APs in the given KG. We encode the input question and the RR-APs via two independent BERT [10] models. Given a question $q$ and an RR-AP $p$, we compute the dot similarity of the

embeddings of corresponding `[CLS]` tokens, i.e.,

$$V_q = \text{BertCLS}_1(q),$$
$$V_p = \text{BertCLS}_2(p), \tag{4}$$
$$\text{sim}(p, q) = (V_q)^T \cdot V_p,$$

where BertCLS denotes the representation of corresponding `[CLS]`. Specifically, we serialize each RR-AP via the corresponding relation labels and a link tag for denoting how the relations are connected. Table 1 illustrated how to serialize RR-APs of different structures.

**Table 1: Serialization of RR-APs.**

| RR-APs | Serialization |
|---|---|
| $\xleftarrow{rel_1} \bigcirc \xrightarrow{rel_2}$ | `[CLS] rel₁ SS rel₂ [SEP]` |
| $\xleftarrow{rel_1} \bigcirc \xleftarrow{rel_2}$ | `[CLS] rel₁ SO rel₂ [SEP]` |
| $\xrightarrow{rel_1} \bigcirc \xrightarrow{rel_2}$ | `[CLS] rel₁ OS rel₂ [SEP]` |
| $\xrightarrow{rel_1} \bigcirc \xleftarrow{rel_2}$ | `[CLS] rel₁ OO rel₂ [SEP]` |

The dense encoders are trained with the cross-entropy loss on the output logits. We use heuristically constructed pseudo EPs to avoid manual annotation. For each question in the training set, we randomly select one of its answers, and subsequently collect 1 or 2 hops paths between the topic entities and the selected answer to construct a pseudo-EP.[1] The RR-APs on the pseudo-EP(s) are considered positive samples. The negative samples are generated in two ways. Half of the negative samples are generated by randomly replacing one relation or the tag of a positive sample, and the others are randomly sampled over the entire KG.

In the test process, we use the dense encoders to retrieve $K$ most relevant RR-APs and collect all the ER-APs of topic entities as candidate APs.

---

**Algorithm 1** The construction of candidate EPs

1: **function** ENUMERATE($\tau, AP_{ER}, AP_{RR}$)
2:    $C \leftarrow \emptyset$
3:    **for** $p \in AP_{ER}$ **do**
4:       $AP'_{ER} \leftarrow AP_{ER} \setminus \{q \in AP_{ER} \mid q.ent = p.ent\}$
5:       $C \leftarrow C \cup \text{ITERExpAND}(p, \tau, AP'_{ER}, AP_{RR})$
6:    **return** $C$

---

## 4.2 Candidate Evidence Pattern Construction

After retrieving the APs, we enumerate all possible EPs with Algorithm 1. The algorithm starts from an ER-AP and iteratively expands the under-construction EP by Algorithm 2 ITERExAPND. ITERExAPND uses a threshold $\tau$ to control the maximum size of EPs. The value of $\tau$ varies with different datasets. Since APs have already recorded the atomic adjacency, the algorithm only needs to check whether the adjacency recorded by an AP is consistent with the current EP. An atomic expansion happens on a variable node. The variable will be replaced by a topic entity (for expansion with

---

[1] CVT connections over entities in Freebase are treated as 1-hop during the collection process. Therefore, the maximum length of relation paths is 4, rather than 2.

---

**Algorithm 2** The iterative expansion of under-construction EPs

1: **function** ITERExPAND($P, \tau, AP_{ER}, AP_{RR}$)
2:    $C \leftarrow \emptyset$
3:    **if** IsVALIDPAT($P$) **then**
4:       $C \leftarrow C \cup \{P\}$
5:    **if** $|P| = \tau$ **then**
6:       **return** $C$
7:    **for** $p \in \{p \in AP_{ER} \mid \text{ExPANDABLE}(P, p)\}$ **do**
8:       $P' \leftarrow \text{ExPAND}(P, p)$
9:       $AP'_{ER} \leftarrow AP_{ER} \setminus \{q \in AP_{ER} \mid q.ent = p.ent\}$
10:      $C' \leftarrow \text{ITERExPAND}(P', \tau, AP'_{ER}, AP_{RR})$
11:      $C \leftarrow C \cup C'$
12:    **for** $p \in \{p \in AP_{RR} \mid \text{ExPANDABLE}(P, p))\}$ **do**
13:      $P' \leftarrow \text{ExPAND}(P, p)$
14:      $C' \leftarrow \text{ITERExPAND}(P', \tau, AP_{ER}, AP_{RR})$
15:      $C \leftarrow C \cup C'$
16:    **return** $C$

---

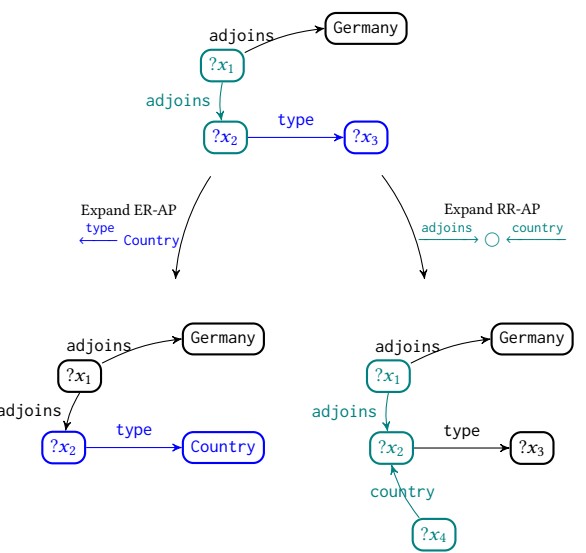

**Figure 4: Expand an under-constructed EP with an ER-AP (the left side) or an RR-AP (the right side).**

ER-AP) or extended with a relation (for expansion with RR-AP), as illustrated in Figure 4.

Specifically, the algorithm assumes that each topic entity can only be expanded once (line 4 of Algorithm 1 and line 9 of Algorithm 2), but the use of RR-APs is not limited. After expansion, the EPs that can provide answers will be considered as candidates (line 3 of Algorithm 2). There are two critical predictive functions, IsVALIDPAT and ExPANDABLE, with the following criteria for their checks:

IsVALIDPAT checks whether the pattern could correspond to the meaning of the question. It requires the pattern to include all topic entities and also demands the existence of knowledge graph nodes that satisfy the variables in the pattern. To ensure the simplicity of the pattern and avoid introducing meaningless relations, we do not

accept an arbitrary expansion of relation paths. Specifically, only two types of structures are considered valid:

- All endpoints of the pattern (i.e., nodes with a degree of 1) are topic entities. In this case, the evidence pattern is a minimal structure that can connect all the topic entities.
- There is only one variable endpoint. We allow this exception in case the correct pattern is a simple triplet or all topic entities describe the query target through a common intermediate.

EXPANDABLE assesses whether a pattern $P$ can be expanded with atomic pattern $p$. The EXPAND process will try to expand $P$ with $p$ to obtain a new EP $P'$. EXPANDABLE return true if only it is possible to construct an expanded $P'$ where all corresponding atomic patterns appear in $AP_{ER}, AP_{RR}$. Taking the expansions in Figure 4 as an example, the expansion with "$\xrightarrow{adjoins} \bigcirc \xleftarrow{country}$" requires the existence of "$\xleftarrow{type} \bigcirc \xleftarrow{country}$".

## 4.3 Evidence Pattern Ranking

The enumeration will generate multiple candidate EPs, and we model the probability $\Pr_\phi$ over candidate EPs via a BERT-implemented cross-encoder. The model design is similar to the query ranking models of the mainstream semantic parsing methods [14, 35]. For a question $q$ and a candidate EP $P$, the process can be formulated as follows:

$$sim(P, q) = \text{LINEAR}(\text{BERTCLS}([q; P])).  \quad (5)$$

We serialize the candidates as sequences of triplets, where resources are denoted by their Freebase labels (including domains of relations). We concatenate the serialization of EP and the corresponding question as the input of the cross-encoder. Sample input for the EP illustrated in Figure 2 is demonstrated as follows:

```
[CLS] what ... border ? [SEP] ?u ...adjoins ?v ; ...
?w ...containedby stadhuis ; ?u ...type country ; [SEP]
```

The model is trained with the cross-entropy loss on the output logits during the training process. we take the candidate EPs that cover the most answers as positive examples, and the others as negative examples.

## 5 EVALUATION

### 5.1 Experimental Settings

*5.1.1 Datasets.* We evaluate our method on two widely used benchmarks over Freebase [3], Complex WebQuestions 1.1 (denoted as CWQ) and WebquestionSP (denoted as WebQSP). The statistics about them are presented in Table 2. The split of validation and training data of WebQSP follows Zhang et al. [38]. We use the latest dump of Freebase[2]. We build a Faiss index with 2,366,590 relation-relation atomic patterns.

*5.1.2 Compared Methods.* We compared with seven two-staged IR-KGQA methods [15, 18, 24, 26, 28, 29, 38] and an in-context learning method **KD-COT** [33]. The two-staged IR-KGQA methods propose five subgraph extraction methods and six answering reasoning methods. The subgraph extraction methods are listed as follows:

[2]https://developers.google.com/freebase

**Table 2: Statistics of the number of questions.**

| Dataset | #Train | #Val. | #Test |
|---|---|---|---|
| CWQ | 27,639 | 3,519 | 3,531 |
| WebQSP | 2,848 | 250 | 1,639 |

- Saxena et al. [26] proposes a *relation-pruning strategy* to extract subgraphs connected by allowed relations. We denote it as **R-Prune**.
- **PPR** denotes the heuristic idea proposed by Sun et al. [29]. They extract a subgraph with *personalized PageRank* scores computed on the neighborhoods of topic entities.
- **PullNet** [28] iteratively constructs question-specific subgraph by "pull" operations on KG and text corpus.
- **SR** denotes the *subgraph retriver* proposed by Zhang et al. [38]. It expands relation paths via a sequential decision process.
- **UniKGQA** [18] unifies subgraph extraction and answer reasoning by computing and propagating matching information between questions and relations.

The answer reasoning methods are listed as follows:

- Miller et al. [24] proposes a *key-value memory network* to store KG facts, which implicitly models 1-hop neighboring graphs of topic entities. We denoted it as **KV-Mem**.
- **EmbedKGQA** [26] formulates answer reasoning as a link prediction task.
- **GCN** denotes the idea to identify answers via *graph convolutional network* [29].
- **NSM** denotes *neural state machine* for KGQA [15]. It iteratively generates instruction vectors and updates the entity distribution to predict the final answer(s).
- **UniKGQA** [18] uses the same architecture for subgraph extraction and answer reasoning.
- **KD-COT** [33] is an in-context learning method. It proposes the *knowledge driven chain-of-thought* reasoning process to iteratively retrieve KG.

We use NSM as the answer reasoner to implement our KGQA system.

In addition to the above IR-KGQA methods, we also provide the results of three recent semantic parsing methods for reference, including two fine-tuning methods **DecAF** [37] and **Program Transfer** [7] and a few-shot method **KB-Coder** [25]. We report the results of these methods results because they can run with and without full annotation of gold queries.

*5.1.3 Implementation Details.* We implemented EPR with Python 3.7 and PyTorch 1.9. The results are obtained on a server with Intel Xeon Gold 5222 CPUs and NVIDIA RTX 3090 GPUs [3]. We use bert-base-uncased as the neural model for atomic pattern retrieval and evidence pattern ranking. The hyper-parameters are presented in Table 3.

The size threshold $\tau$ of evidence patterns is decided by the sizes of collected pseudo EPs. Specifically, the thresholds for CWQ and WebQSP are set to 5 and 3 respectively. We grid-search the threshold

[3]The implementation will be open-sourced upon acceptance for publication.

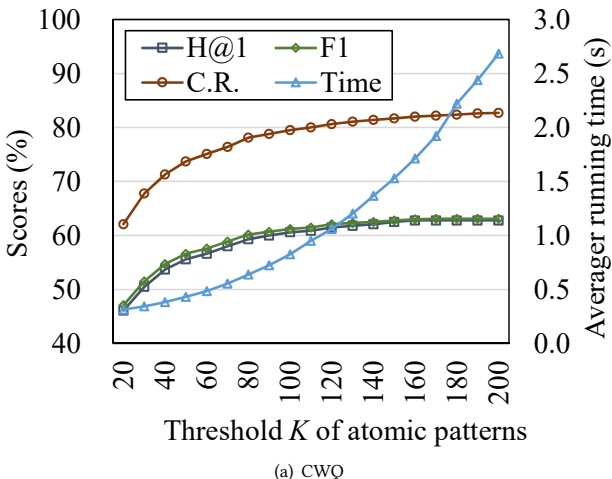

(a) CWQ

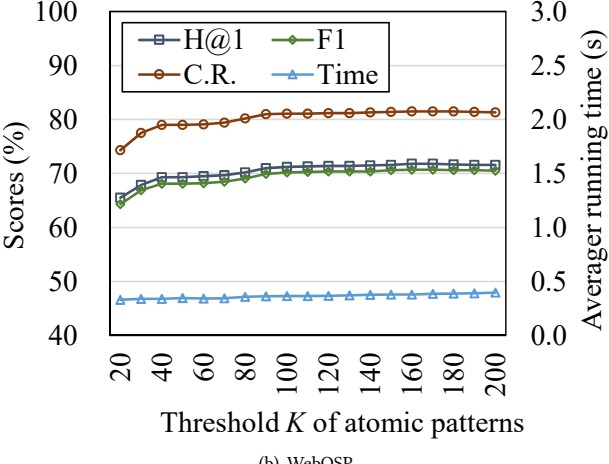

(b) WebQSP

**Figure 5: The performance and execution time of EPR+NSM with various numbers of APs on CWQ (a) and WebQSP (b).**

**Table 3: Hyper-parameters for the models.**

| Parameters | AP retrieval | EP ranking |
|---|---|---|
| Vector dimension | 768 | 768 |
| Batch size | 16 | 2 |
| Epochs | 5 | 10 |
| Initial learning rate | 2e-5 | 1e-5 |

$K$ for the maximum number of retrieved RR-APs from 20 to 200 with a step of 10. Larger $K$ may increase the running time while improving the answer cover rates.

*5.1.4 Evaluation Metrics.* We report Hits@1 (denoted as H@1), F1 score, and answer cover rate (denoted as C.R.) of compared methods. Hits@1 directly evaluates whether the top-1 predicted answer is correct. Because some questions have multiple answers, we also use the F1 score to evaluate the system outputs. The predicted answers of our system are truncated according to the default threshold of the answer reasoner. The computation of C.R. follows Zhang et al. [38], which is the proportion of questions for which the extracted subgraph contains at least one answer. Specifically, C.R. can be regarded as the H@1 with an oracle answer reasoner. It reflects the performance of subgraph extraction and helps to identify the performance of each module separately.

## 5.2 Main Results

Table 4 reports the main results of compared methods. Due to space limitations, the table only includes our implementation with the top 100 retrieved RR-APs. The comprehensive set of results of our implementation is illustrated in Figure 5 and is discussed in Section 5.3.

Our implementation has achieved a new SOTA for IR methods on the ComplexWebQuestions (CWQ) dataset. Our implementation shows significant improvements, with a +4.9 increase in H@1 (compared to KD-COT) and a +13.2 increase in F1 score (compared to

**Table 4: The evaluation results(%) on CWQ. The best results of IR methods are in bold, and the second-best results are underlined. † denotes that the method requires gold query annotation of all training questions. ∗ denotes few-shot methods.**

| Method | CWQ | | WebQSP | |
|---|---|---|---|---|
| | H@1 | F1 | H@1 | F1 |
| *Semantic Paring Methods* | | | | |
| DecAF w/o Gold Query [37] | 50.5 | - | 74.7 | 49.8 |
| †DecAF w/ Gold Query [37] | 68.1 | - | 80.7 | 77.1 |
| Program Transfer [7] | 58.1 | 58.7 | 74.6 | 76.5 |
| *KB-Coder [25] | - | - | - | 60.5 |
| †*KB-Coder + Retrieval [25] | - | - | - | 75.2 |
| *Information Retrieval Methods* | | | | |
| KV-MeM [24] | 18.4 | 15.7 | 46.6 | 34.5 |
| R-Prune + EmbedKGQA[26] | 32.0 | - | 66.6 | - |
| PPR + GCN [29] | 36.8 | 32.7 | 66.4 | 60.4 |
| PullNet + GCN [28] | 45.9 | - | 68.1 | - |
| PPR + NSM [15] | 47.6 | 42.4 | 68.5 | 62.8 |
| SR + GCN [38] | 49.0 | 42.7 | 66.7 | 63.1 |
| SR + NSM [38] | 50.2 | 47.1 | 69.5 | 64.1 |
| UniKGQA + NSM [18] | 49.2 | - | 69.1 | - |
| UniKGQA + UniKGQA [18] | 50.7 | 48.0 | **75.1** | 70.2 |
| *KD-COT [33] | 55.7 | - | 68.6 | 52.5 |
| EPR$_{K=100}$ + NSM (OURS) | **60.6** | **61.2** | 71.2 | 70.2 |

UniKGQA). On the WebQSP dataset, our implementation exhibits competitive performance compared to the SOTA method UniKGQA. There is a decrease in H@1 by -3.9 points, but the F1 score remains similar. Moreover, our implementation surpasses other NSM-based methods. Specifically, our implementation demonstrates notable

improvements, with a +1.7 increase in H@1 and a +6.1 increase in F1 compared to other NSM-based methods.

When considering all methods that do not require complete gold query annotations in the training data, our implementation outperforms the current SOTA method, Program Transfer, on the CWQ dataset. The improvements are evident, with a +2.5 increase in F1. However, the results on the WebQSP dataset are lower, showing a -6.3 decrease in F1.

The result reported in Figure 5 demonstrated that the ratio of Hits@1 to answer cover rates (i.e., the performance of NSM) is relatively stable. The specific results are reported in Table 5. The average ratios of Hits@1 to C.R. are approximately 76% and 88% on CWQ and WebQSP, respectively. For reference, SR+NSM with a coverage rate threshold of 0.8 only achieves Hits@1 scores below 0.4 on CWQ and below 0.65 on WebQSP [38]. Despite the varying objectives of different subgraph extraction methods making the ratios not immediately comparable, the significant gap suggests that EPR may offer advantages for downstream reasoning tasks.

**Table 5: The average Hits@1(%), answer cover rates(%) and their ratio over different threshold $K$. The last column reports the average ratios with the standard deviations in brackets.**

| Dataset | avg. H@1 | avg. C.R. | avg. H@1/C.R. |
|---------|----------|-----------|---------------|
| CWQ     | 59.1     | 77.9      | 75.9 (6.4e-3) |
| WebQSP  | 70.5     | 80.2      | 87.9 (1.8e-3) |

## 5.3 Results with Various Number of Retrieved Atomic Patterns

This section discusses the influence of the number of retrieved atomic patterns (APs) on the performance and efficiency of our implementation. We computed the average running time and the performance scores under different thresholds of APs, as illustrated in Figure 5. Our further analysis shows that the increase in running time on CWQ is sorely due to the combinatorial explosion of candidate evidence patterns, as illustrated in Figure 6. In conclusion, the results indicate a notable impact of the number of APs on complex questions (i.e. CWQ), but the difference it brings is not very significant on relatively simple questions (i.e. WebQSP). Specifically, on CWQ, the range of answer cover rates is from 62.1% to 82.7%, and the range of time is from 0.31s to 2.68s. On WebQSP, the range of answer cover rates is from 74.3% to 81.3%, and the range of time is from 0.33s to 0.40s. For complex questions, increases in atomic patterns bring efficiency bottlenecks, but the performance improvement brought by increasing the number is diminishing. For relatively simple questions, the retrieved atomic patterns do not have too many combinations, and the increase in the number threshold has few effects on the system.

## 5.4 The Impact of Training Data Size on Pattern Ranking

We conducted an experiment to assess the influence of training data size on the ranking of candidate evidence patterns. We randomly split the training data into five equal parts and trained the ranking

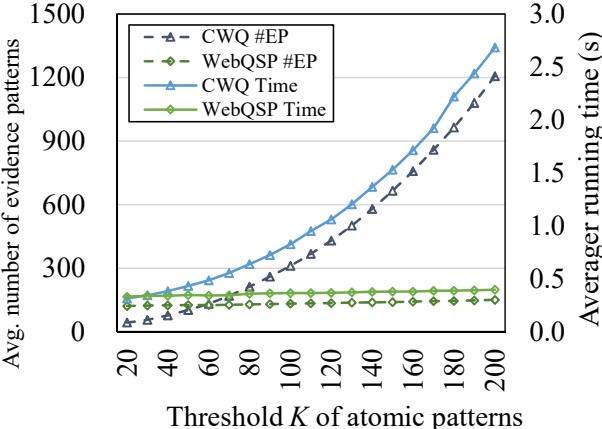

**Figure 6: The average running time and number of candidate evidence patterns.**

model with various ratios of data, as reported in Table 6. The results indicate that the system's performance is not particularly sensitive to this. We suppose this is because the retrieval of atomic patterns already establishes the question-related KG context, and the ranking model primarily focuses on structural information, with less reliance on data size.

**Table 6: The results (%) obtained by training the ranking model with various ratios of training data. All results are obtained with $K = 100$.**

| Ratio | CWQ | | | WebQSP | | |
|-------|------|------|------|------|------|------|
|       | C.R. | H@1  | F1   | C.R. | H@1  | F1   |
| 20%   | 77.7 | 59.1 | 59.6 | 75.5 | 65.9 | 64.0 |
| 40%   | 79.4 | 59.8 | 60.6 | 78.5 | 68.3 | 67.8 |
| 60%   | 79.1 | 59.8 | 60.4 | 79.4 | 68.5 | 67.8 |
| 80%   | 78.7 | 59.5 | 60.5 | 81.4 | 70.3 | 69.6 |
| Full  | 79.5 | 60.6 | 61.2 | 81.1 | 71.2 | 70.2 |

## 5.5 Error Analysis

We conduct an error analysis on a sample of 100 questions with incorrect answer predictions obtained from our experiments on CWQ and WebQSP datasets. The analysis revealed various issues contributing to the incorrect answer predictions. We classify the main issues into six types and report the statistics in Table 7. Notably, 20% of the errors on CWQ and 28% of the errors on WebQSP stemmed from EPR failing to cover the correct answers. This issue was primarily linked to the insufficiency of retrieved atomic patterns. A significant proportion of the errors, comprising 54% on CWQ and 30% on WebQSP, were caused by non-entity descriptions of the answers. Many of these descriptions involved numerical reasoning, such as expressions like "higher than 590". This issue is unlikely to be resolved within the current IR-KGQA framework. Even if we extend evidence patterns to capture numerical patterns, the current answer reasoning methods encounter challenges in

handling numerical information. The construction and ranking of evidence patterns contribute to 10% of the errors on CWQ and 28% of the errors on WebQSP. Only a small proportion of errors on CWQ, 8%, were caused by imperfect answer reasoning. This observation indicates EPR's effectiveness in reducing the impact of noisy extraction. Lastly, it is important to note that some of the data has quality issues. For example, our system predicted EgyptianArabic as the top answer to the question "What kind of language do Egyptians speak?", which, while seemingly correct, was not part of the annotated answer set.

**Table 7: The statistics about the main issues of incorrect answer prediction.**

| Main issues | CWQ | WebQSP |
| --- | --- | --- |
| Insufficient AP | 20% | 28% |
| EP construction errors | 6% | 8% |
| EP ranking errors | 4% | 20% |
| Non-entity evidence | 54% | 30% |
| Answer reasoning errors | 8% | 0% |
| Data quality | 8% | 14% |

Besides, our further analysis shows that unseen relation is a crucial issue for our implementation. About 2.8% (98/3531) of CWQ test questions and 5% (83/1639) of WebQSP test questions contain relations that never appear on the training questions. Our implementation experiences a sharp decline on these questions, as illustrated in table 8.

**Table 8: The results (%) on questions with and without unseen relations. All results are obtained with $K = 100$.**

| Unseen Rel. | CWQ | | | WebQSP | | |
| --- | --- | --- | --- | --- | --- | --- |
| | C.R. | H@1 | F1 | C.R. | H@1 | F1 |
| with | 36.7 | 17.4 | 19.7 | 49.4 | 36.1 | 35.4 |
| w/o | 80.7 | 61.8 | 62.4 | 82.8 | 73.1 | 72.0 |

## 6 CONCLUSION

In this paper, we propose evidence pattern retrieval (EPR), which aims to improve the subgraph extraction of IR-KGQA methods by reducing noisy facts. Our main contribution can be summarized as follows:

- We propose the novel idea of evidence pattern, which refers to how necessary resources (entities and relations) are connected to support a knowledge graph node as an answer to a question. It enables the explicit modeling of structural dependencies during the subgraph extraction process of IR-KGQA.
- We propose an efficient implementation of EPR. It takes an evidence pattern as a combinations of the atomic adjacency patterns of resource pairs. We build a vector index for fast retrieval of the atomic patterns and propose an expanding algorithm to construct evidence patterns.

- We evaluate the EPR-based KGQA system with a rich experimental analysis. Our analysis demonstrates the importance of structural dependencies and EPR's ability to handle complex questions.

Although EPR significantly enhances IR-KGQA methods in handling complex questions, there are still issues worth further exploration. In this paper, we implement atomic pattern retrieval using a BERT-based bi-encoder. Experimental results indicate that its performance is suboptimal in cases where the retrieval threshold is low (e.g., $\leq 40$) or where unseen relations are present in questions. It is necessary to explore solutions that perform better without significantly compromising retrieval efficiency. Our implementation on CWQ leads to a combinatorial explosion as the retrieval threshold increases, and it may require necessary optimizations for the brute-force enumeration of EP to improve efficiency. Exploring the combination of EPR and in-context learning to achieve state-of-the-art performance in a few-shot manner is worth investigating. Besides, current IR-KGQA methods lack the ability to model numerical information. The possibility of modeling numerical features as pattern information to enhance the downstream answer reasoning methods' capability is also worth investigating.

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
