# OpenReview forum: "Enhancing Complex Question Answering over Knowledge Graphs through Evidence Pattern Retrieval"
_ACM.org/TheWebConf/2024/Conference — TheWebConf24_

### Official Review · Reviewer_kpRQ · 2023-11-19

**Novelty:** 3
**Technical Quality:** 5

**Review:**

Summary:

In two-stage IR-based KGQA systems, the quality of the subgraph plays a pivotal role in determining QA performance. This paper emphasizes subgraph extraction. The authors introduce the EPR method, which initiates the process by extracting atomic patterns through dense retrieval. Subsequently, it systematically enumerates combinations to generate candidate evidence patterns, ultimately selecting the top-ranked pattern based on inferences from a trained model. The experimental results show that on CWQ, EPR + NSM achieves the best performance at H@1 and F1. However, on WebQSP, there is still a performance gap at H@1. For certain issues raised in the paper, the authors have provided detailed explanations. Overall, I think the paper is well-written and conducts detailed experiments to demonstrate the effectiveness of the proposed method.

Pros:
+ Since in the two-stage IR-based KGQA systems, the quality of the subgraph is crucial and decisive to the QA performance, the paper focuses on the interesting subgraph extraction.
+ The paper proposed the method of subgraph construction that gradually expands the evidence path and provides explainability for answering the questions.
+ The authors conducted extensive experiments and proved the proposed subgraph construction method is effective.

Cons:
- In some subsections, such as 4.1, it would be better to add items to highlight different topics, e.g., the RR-APs dense encoder is different from atomic pattern retrieval, which makes the paper more readable.
- The paper lacks an explanation of some technical details or the expression is not very clear, e.g., the two BERT-based models (dense encoder and EP ranking).
- Freebase KG is a relatively small and old KG compared with Wikidata, which is much larger and more popular. If the KG is large, the cost is large because it needs to pre-compute the dense encoder for building the index for retrieving candidate RR-APs.

**Questions:**

* In the Figure 3, the initial evidence facts (ER-APs) of the question "What country, containing Stahuis, does Germany border?" contain "Country", but in general, the word "country" isn't annotated as the topic entity. If the system starts from the topic entities, why is the node "country" involved in the initial facts? Does the system also takes care of the answer type and takes it into consider when extracting facts from KG?
* When training the dense encoders and sampling the positive and negative samples, what is the ratio of positive and negative samples? The paper does not mention it. The ratio is an important factor/parameter for the performance of dense retriever.
* How do the authors deal with the relations in CVT when the CVTs are treated as 1-hop fact? Are they combined as one relation, e.g. spouse + start time?
* In the test process, the authors use the dense encoders to retrieve K most relevent RR-APs and then collect ER-APs of the topic entities. In the process, is there the case that in the top-K RR-APs, no RR-APs are related to one or some of the topic entities when there are multiple topic entities in the question so that the system fails to find ER-APs.
* In the BERT-implemented cross-encoder for evidence pattern ranking, how to construct training data? How is the efficiency or running time for training and inferencing?
* When sampling negative samples for training RR-APs dense encoder, the author mentioned some of samples are randomly sampled over the entire KG. Would you please explain how to sample over the entire KG? Did you first construct RR-APs for the entire KG and then sample from RR-APs of the KG?

**Reviewer Confidence:**

4: The reviewer is certain that the evaluation is correct and very familiar with the relevant literature

**Scope:**

4: The work is relevant to the Web and to the track, and is of broad interest to the community

---

### Official Review · Reviewer_25Kx · 2023-11-22

**Novelty:** 5
**Technical Quality:** 6

**Review:**

The paper aims at improving the results of current Information Retrieval methods for question answering over knowledge graphs (ID-KGQA). In particular, authors focus on reducing the potential noisy facts in the knowledge graph when selecting a candidate subgraph to later perform the answer reasoning tasks. While several mechanisms and heuristics exist (e.g. PageRank), the authors propose to capture the structure of the graph that leads to the final answers, referred to as evidence pattern. The authors propose to use basic atomic patterns that can be combined/expanded and ranked (authors use a  BERT cross-encoder). The practical implementation (using some fast indexes) and evaluation shows that this subgraph selection improves existing methods on complex queries.

The paper proposal and evaluation is technically sound and complete. I particularly appreciate the attention to the reproducibility (algorithms are provided, and source code will be made available) and the evaluation result details.

**Questions:**

- In KGs such as in RDF, there could be a logical relationship between two relationships (e.g. a "father" relationship could be a subproperty of "relationship" or, "husband of" is the opposite of "wife of") or entities (A house is a type of Building). I understand that this is treated as another relationship if the semantics is present in the KG, but perhaps authors can specify a bit more if they consider any particularities of this semantics (TBox).

- Which of the features of well know query languages, e.g. SPARQL, cannot be supported with such simple patterns? Authors mentioned the impossibility to model numerical information, but there could be others (e.g. OPTIONAL, GROUP BY)

Edit: I thank authors for their clarifications

**Ethics Review Description:**

no issues

**Reviewer Confidence:**

2: The reviewer is willing to defend the evaluation, but it is likely that the reviewer did not understand parts of the paper

**Scope:**

4: The work is relevant to the Web and to the track, and is of broad interest to the community

---

### Official Review · Reviewer_7jSt · 2023-11-23

**Novelty:** 4
**Technical Quality:** 5

**Review:**

The paper deals with the problem of question answering over knowledge graphs. The proposed method tries to build a subgraph containing the desired answer by iteratively constructing and expanding evidence subgraphs using atomic graph patterns (such as property-object or incoming property-object-outgoing property). After constructing candidate patterns, a trained ranking model utilizing BERT is used to rank them according to their relevance to the question. The algorithm is evaluated on top of two benchmark datasets and compared against a set of both semantic parsing and information retrieval methods, showing an improvement on the CWQ benchmark and on-par performance with the UniKGQA algorithm on the WebQSP one.

The natural language question answering over knowledge graphs is a well-studied topic in the community and a relevant one for the conference and the track. While the overall approach of trying to find the answer by combining atomic subgraph patterns has been used in the past, the proposed techniques for the subgraph expansion and candidate subgraphs ranking look interesting and seem to result in corresponding improvements in performance.

One comment regarding the choice of methods to compare: these methods include several IR-based and SP-based methods originated in the research community. However, given the recent rise in LLM-based models, wouldn’t it be helpful to include a fine-tuned extension of a publicly available LLM model (such as LLaMA) as a “naïve baseline” to demonstrate the added value of dedicated QA methods?

**Questions:**

- How would the approach compare with fine-tuning public general-purpose LLM models?

**Reviewer Confidence:**

1: The reviewer's evaluation is an educated guess

**Scope:**

3: The work is somewhat relevant to the Web and to the track, and is of narrow interest to a sub-community

---

### Official Review · Reviewer_Goiu · 2023-11-26

**Novelty:** 4
**Technical Quality:** 6

**Review:**

This paper proposes a new method for extracting subgraphs for Information Retrieval style question answering over knowledge graphs. The approach focuses on extracting structural dependencies between entities and relations. Some of the key ideas include two types of atomic patterns, pattern expansion and ranking. The authors compared the proposed method to several SOTA approaches, where the proposed EPR method shows noticeably better scores in most settings (except for H@1 for WebQSP). The error analysis was also helpful in understanding where future improvements could be focused on. Generally speaking, this is an interesting work and also demonstrates good results on benchmark datasets.

Pros:
1) Interesting idea of extracting the structural dependencies for subgraph extraction and then QA.
2) Good experiments on two benchmark datasets that demonstrate mostly better performance than several SOTA systems.
3) The error analysis is helpful as it shows some potential future directions.

I have a few questions to be listed in the Questions section.

**Questions:**

1) I understand that the authors focus on atomic patterns in this paper. I am simply curious about the trade-offs between limiting to simpler patterns vs. the capability of answering more complex questions. For example, would considering more complex patterns allow correctly answering some complex questions that the current approach failed to?

2) Figure 5 shows the impact of different K values. It seems that the choice of K has a more evident impact on the CWQ dataset. For example, after K=40, there is only some minor improvements on WebQSP while we still observe good improvements on CWQ. Is this due to the complexity of the questions or some other reason?

3) A related question for Table 6: Apparently, leveraging more training data helped improvement performance on WebQSP but we don't see similar impact on CWQ (some minor to slight improvements). Still, I am wondering whether this is due to the complexity of the questions. For example, the questions in CWQ are more complex; thus, even with more training data, the system simply couldn't learn more on this dataset.

**Reviewer Confidence:**

3: The reviewer is confident but not certain that the evaluation is correct

**Scope:**

4: The work is relevant to the Web and to the track, and is of broad interest to the community

---

### Decision · Program_Chairs · 2024-01-22

**Decision:**

Accept

**Comment:**

The paper presents a novel method, EPR, for subgraph extraction in Information Retrieval style question answering over knowledge graphs. Reviewers commend the interesting approach, good experimental results, and thorough error analysis. Questions raised by reviewers mainly focus on trade-offs between simple and complex patterns, the impact of parameter choices on different datasets, and the comparison with fine-tuned language model baselines. There are also inquiries about handling semantic relationships in RDF and the consideration of various query language features. The technical quality is generally praised, but some reviewers suggest improvements in paper organization and additional clarification of technical details. Overall, the work is deemed relevant, novel, and technically sound, with valuable contributions to the community.

 The authors' have addressed the identified weaknesses in their rebuttal in a satisfactory manner and proposed ways to address those in the CR.

 In the light of the above, I suggest accepting this paper.